

# An application of PCR-RFLP species identification assay for environmental DNA detection

Takeshi Igawa[1], Teruhiko Takahara[2,3], Quintin Lau[4] and Shohei Komaki[5]

[1] Amphibian Research Center, Hiroshima University, Higashi-Hiroshima, Hiroshima, Japan
[2] Faculty of Life and Environmental Science, Shimane University, Matsue, Shimane, Japan
[3] Estuary Research Center, Shimane University, Matsue, Shimane, Japan
[4] Department of Evolutionary Studies of Biosystems, Sokendai (The Graduate University for Advanced Studies), Hayama, Kanagawa, Japan
[5] Division of Biomedical Information Analysis, Iwate Tohoku Medical Megabank Organization, Disaster Reconstruction Center, Iwate Medical University, Shiwa, Iwate, Japan

Corresponding authors
Takeshi Igawa,
tigawa@hiroshima-u.ac.jp
Teruhiko Takahara,
ttakahara@life.shimane-u.ac.jp

## ABSTRACT

Recent advancement of environmental DNA (eDNA) methods for surveying species in aquatic ecosystems has been used for various organisms and contributed to monitoring and conservation of species and environments. Amphibians are one of the promising taxa which could be monitored efficiently by applying quantitative PCR (qPCR) or next generation sequencing to eDNA. However, the cost of eDNA detection using these approaches can be quite high and requires instruments that are not usually installed in ecology laboratories. For aiding researchers in starting eDNA studies of amphibians, especially those not specialized in molecular biology, we developed a cost efficient protocol using PCR-RFLP method. We attempted to detect eDNA of three Japanese *Rana* species (*Rana japonica, Rana ornativentris*, and *Rana tagoi tagoi*) in various spatial scales including an area close to the Fukushima nuclear power plant where the environment is recovering after the disaster in 2011. Our PCR-RFLP protocol was successful in detecting *Rana* species in static water in both laboratory and field; however, it could not detect *Rana* species in non-static water samples from the field. Even a more sensitive detection method (standard qPCR) was unable to detect frogs in all non-static water samples. We speculate that our new protocol is effective for frogs living in lentic habitats, but not for lotic habitats which may still require the gold standard of field observation for detection approach.

## INTRODUCTION

Spatial distribution of organisms is one of the key components for understanding the ecology of the species and the ecosystem in which they inhabit. In addition, monitoring of species and population distribution are essential for their conservation. However, traditional visual surveys and monitoring techniques can be problematic, related to seasonal, ecological, and ethological differences within a target species, and difficulties in

distinguishing between closely related sympatric species and varied life stages. In recent studies, environmental DNA (eDNA) methods, which involve the detection of species-specific DNA fragments in the environment, have been used for surveying species in aquatic ecosystems (*Takahara, Minamoto & Doi, 2013*; *Rees et al., 2014*; *Goldberg, Strickler & Pilliod, 2015*). Although there are some issues related to field surveys, this is a promising and non-invasive method, and thus has been used for various organisms including both plants and animals in terrestrial and aquatic environments (*Thomsen & Willerslev, 2015*).

However, such eDNA detection experiments usually require expensive instruments and reagents for quantitative PCR (qPCR) (for single species) or next generation sequencing (NGS) (for multiple species). This potentially limits the number of institutions able to conduct eDNA studies. For sustainable monitoring activities and conservation of species diversity, an efficient and easy method should be developed for many laboratories close to a field site. Especially for amphibians, several studies support that detection is more cost-effective with eDNA than with traditional monitoring approaches (*Olson, Briggler & Williams, 2012*; *Pilliod et al., 2013*; *Biggs et al., 2015*; *Buxton et al., 2017*). Amphibians are also intrinsic target species for eDNA studies in terms of their ecology because they have both aquatic and terrestrial life cycles and inhabit different kinds of water bodies, such as river (lotic) and pond (lentic) water systems, depending on species and life stages.

Although the sensitivity and cost of detection methods could be correlated, applying the low cost PCR-RFLP for eDNA detection has already been successful in differentiating several salmonid fish species (*Clusa et al., 2017*). However, to-date this PCR-RFLP approach has not been applied to organisms that are not fully aquatic, including amphibians. Therefore, it would be important to develop a cost efficient protocol using PCR-RFLP method for detection of eDNA fragments, instead of qPCR, in amphibian species.

In this study, we attempted to develop a protocol for detecting the presence of three true frog species from the *Rana* genus (*Rana japonica*, *Rana ornativentris* and *Rana tagoi tagoi*). These species are distributed across mainland Japan, including Honshu, Shikoku, and Kyushu islands. *Rana japonica* and *Rana ornativentris* are often located near areas of human development and thus could be threatened with population decline due to human-related habitat loss. As a result, these two species have been listed as locally threatened in some regions (*Natural Environment Division & Bureau of Environment TMG, 2010*; *Natural Environment Division & Department of Environmental Affairs FPG, 2014*). In addition, these species are key indicator species for assessing the degree of disturbance to the natural environment, especially in Japanese traditional rural areas called "Satoyama" (*Kitagawa, 2002*). For example, agencies like the Biodiversity Center of Japan, the Nature Conservation Bureau, and the Ministry of the Environment (e.g., Monitoring-Site 1000 project) have conducted ecological monitoring surveys and listed the occurrence of *Rana* species in numerous reports. Habitat selection of the three species differs to some extent, whereby each species inhabits a wide range of microhabitats. *Rana japonica* and *Rana ornativentris* are mainly found in grasslands that range from lowlands to hillsides; they migrate to marshes, ponds, or paddy fields before the breeding season

in winter (*Maeda & Matsui, 1999*), but *Rana ornativentris* tends to dwell at higher elevations than *Rana japonica* (*Kuramoto & Ishikawa, 2000*; *Osawa & Katsuno, 2001*). Another brown frog species, *Rana tagoi tagoi*, resembles *Rana ornativentris* in morphology and is distributed in hilly areas, sometimes in sympatry with the other two species. However, *Rana tagoi tagoi* lay their eggs under the ground on the edges of small mountain streams during spring.

For development of our new protocol in this study, we adapted the PCR-RFLP method that recognizes these three *Rana* species (*Igawa et al., 2015*) for eDNA detection. In this method, mitochondrial 16S rRNA fragments were amplified and digested with two restriction enzymes (SpeI and HphI) to identify species within the samples by electrophoretic banding pattern of the digested fragments. To evaluate the performance of our protocol, we attempted to detect eDNA diluted in water samples and compared the results with visual observation and qPCR methods. To test applicability in various spatial scales, various water samples were collected from regional levels to microhabitat in addition to water tank experiments. In particular, we obtained regional samples covering 60 km from Fukushima nuclear power plant (FNPP) in spring and summer seasons and assessed presence information of *Rana tagoi tagoi* in habitats recovering after the disaster in 2011. Finally, we reported the effectiveness of the protocol for distribution surveys in the field.

# MATERIALS AND METHODS

## Water tank experiment

To characterize basic performance of our PCR-RFLP assay for eDNA detection, we firstly conducted model experiments by preparing eight different water tank culture setups with the following animals: (1) 40 *Rana japonica* tadpoles for 2 days, (2) 40 *Rana japonica* tadpoles for 5 days, (3) nine *Rana japonica* and one *Rana ornativentris* adults for 1 day, (4) one *Rana japonica* and nine *Rana ornativentris* adults for 1 day, (5) five *Rana japonica* and five *Rana ornativentris* adults for 1 day, (6) one *Rana tagoi tagoi* and nine *Rana japonica* adults for 1 day, (7) one *Rana tagoi tagoi* and nine *Rana ornativentris* adults for 1 day, and (8) negative control (to check the specificity of our method)—no animals for 1 day (but water was used previously for other frog species (*Rhacophorus arboreus*)). The tadpoles of *Rana japonica* were collected from a pond in Hiroshima University campus in Higashi-Hiroshima city (Hiroshima Prefecture) in April 2016 just before the experiment and were classified as growth stage 33 according to *Tahara (1974)* (equivalent to stage 37 in *Gosner (1960)*). The adult frogs of *Rana japonica, Rana ornativentris*, and *Rana tagoi tagoi* were collected from Higashi-Hiroshima City in 2014, Kita-Hiroshima-cho (Hiroshima Prefecture) in 2015, and Kisa, Miyoshi City (Hiroshima Prefecture) in 2011, respectively, and reared in the Amphibian Research Center, Hiroshima University, until the start of experiments. We transferred and kept animals in plastic tanks (450 × 650 ×150 mm) filled with 8 L aged tap water (water level about 30 mm which immersed half body of adults). The animals were not fed during the experiment and were held at room temperature around 18 °C under 12 h:12 h light-dark cycle. We collected a 500 mL water sample from each tank after the specific periods
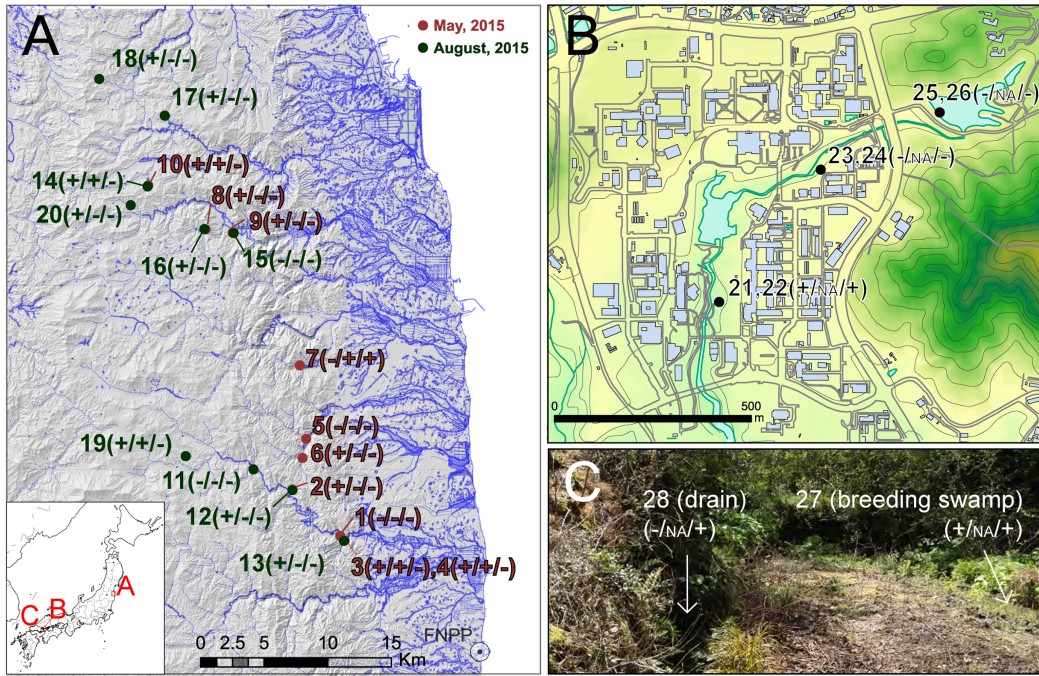

**Figure 1 Sampling localities of field experiments.** Sampling localities of field experiments: (A) regional sampling in Fukushima, (B) local area sampling in Hiroshima University, (C) microhabitat level sampling in Etajima Island. Results of field observation, qRCR and PCR-RFLP detection are indicated, respectively (X/X/X). +, detected; −, not detected; NA, qPCR not conducted. Photo Credit: Takeshi Igawa. Maps were projected by using ArcGIS 9.3 based on a 10-m grid digital elevation map, water line and building outline provided by The Geospatial Information Authority of Japan.

according to the treatments using DNA-free 500 mL bottles, and samples were immediately transferred to a −30 °C freezer. These experiments were conducted in April 2016 for tadpoles (tanks 1 and 2) and September 2016 for adults (tanks 3–8); all animals were returned to the original habitat or enclosures under healthy conditions soon after the experiments.

All procedures were approved by the Hiroshima University Animal Research Committee (Approval number: G17-4) and were carried out in accordance with the recommendations in the Guide for the Care and Use of Laboratory Animals of the Hiroshima University Animal Research Committee.

## Field survey and experiments

To check the performance and sensitivity of PCR-RFLP assay for eDNA from the field environment, we collected water samples from ponds and rivers at 28 sites (Fig. 1). For validation of applicability to monitoring studies in various spatial scales, we collected samples in three hierarchical levels: (A) regional samples covering 60 km from FNPP (sample no. 1–20), (B) local area samples from a single body of water in Hiroshima University campus, Higashi-Hiroshima (sample no. 21–26), and (C) microhabitat level samples from a single breeding site of *Rana japonica* in Etajima Island, Hiroshima Prefecture (N 34.27067, E 132.47679) (sample no. 27 and 28). We collected 500 mL by filling sodium hypochlorite-treated 500 mL bottles (i.e., DNA-free bottles). Field survey

was conducted in spring and summer because the true *Rana* species are active in these seasons. Samples from Fukushima (A) were collected during two periods, May 12, 2015 to May 14, 2015 and August 11, 2015 to August 13, 2015. Samples from Hiroshima University (B) and Etajima (C) were collected on 11 April and 17 April in 2016, respectively. To prevent contamination by a field sampler, we began collecting at downstream sites, and moved upstream as subsequent samples were collected. Sampled water bottles were transported on ice in a cooling box to the laboratory and stored at −30 °C. The subsequent procedure of filtering water samples that were stored frozen was performed within 3 weeks.

In addition, we recorded the presence and absence of frogs and/or tadpoles by visual observation, exploring the vicinity of the field sampling sites. Basically, three or four people conducted field observation for more than 20 min at each site. For sampling in Fukushima, permission to enter the survey area was obtained from the local governments (Iitate Vilage (approval number: 2705-01 and 2707-0) and Namine Town (approval number: 755 and 1959)). Notification of the field survey was accepted by the Iwaki District Forest Office, and permits were obtained from private landowners as required for each site.

## DNA extraction and PCRs

To avoid contamination, we followed a unidirectional lab flow, whereby we performed all PCR protocols, including preparation/addition of the standards and qPCR cycling, in two separate rooms (rooms 1 and 2, respectively). To prevent carry-over contamination, no equipment or samples were returned from room 2 to room 1. In both rooms, laboratory benches were decontaminated using commercial bleach.

The water samples stored at −30 °C were thawed and vacuum-filtered through glass microfiber filters (GF/F; GE Healthcare Bio-Sciences, Pittsburgh, PA, USA) with 0.7 μm mesh size, which were used in other amphibian studies (*Katano et al., 2017*; *Iwai, Yasumiba & Takahara, 2019*). Filter funnels and tweezers used in the filtration treatment were sterilized with 10% commercial bleach (ca. 0.6% hypochlorous acid) for 5 min (i.e., sodium hypochlorite treatment), flushed with a large amount of tap water, and then rinsed with DNA-free distilled water between samples to avoid cross contamination. Filtering controls (i.e., 500 mL of distilled water) in laboratory experiment controls filtered on each day of sample filtration. The filter papers were wrapped in new aluminum foil (i.e., DNA-free), placed in plastic bags, and stored at −30 °C. The eDNA was extracted from the filters according to the methods of *Uchii, Doi & Minamoto (2016)*, using a Salivette tube (Sarstedt, Nümbrecht, Germany) and a DNeasy Blood and Tissue Kit for DNA purification (Qiagen, Hilden, Germany). The filters were incubated by submersion in a mixed buffer (400 μL buffer AL and 40 μL Proteinase K; Qiagen) using a Salivette tube in a dry oven at 56 °C for 30 min. The tubes with filters were centrifuged at 5,000×$g$ for 5 min at room temperature. Then, 220 μL of TE (Tris–EDTA) buffer (pH: 8.0; 10 mM Tris–HCl and one mM EDTA) was added to the filters, and tubes were centrifuged again at 5,000×$g$ for 5 min. Buffer AL (200 μL) and 100% ethanol (600 μL) were then added to each filtrate and mixed by pipetting. The mixture was applied to a DNeasy Mini spin column and centrifuged at 6,000×$g$ for 1 min. This step was repeated until the mixture was completely

processed. We followed the manufacturer's instructions for further steps, and eDNA was eluted from each sample solution with a final volume of 100 µL Buffer AE.

The eDNA samples were then used for PCR-RFLP assay to detect DNA from the three target Japanese brown frog species based on *Igawa et al. (2015)*. This method utilizes species-specific restriction enzyme (SpeI and HphI) digestion sites in a partial nucleotide fragment of mitochondrial 16S rRNA amplified by PCR. However, the oligonucleotide primers used by *Igawa et al. (2015)* could amplify 16S rRNA fragments originating from other vertebrate species including humans which showed similar banding patterns. Therefore, we altered the primers to amplify a shorter region in which only the three target frog species have restriction sites. Specifically, we modified the method of *Igawa et al. (2015)* by changing the primers: F96 5′-GTCCAGCCTGCCCAGTGAYAAA-3′ and R19 5′-GTTGAACAAACGAACCATTGGT-3′. These new primers were designed from an internal region of the previous study and amplify shorter fragments (*Rana japonica*: 514–517 bp, *Rana ornativentris*: 516–517 bp, and *Rana tagoi tagoi*: 514–516 bp).

For higher efficiency of PCR amplification, we also modified the PCR protocol described by *Igawa et al. (2015)*. In this study, PCR was conducted using KOD FX Neo (TOYOBO, Osaka, Japan). The 20 µL total volume of reaction solution included 10 µL of 2 × PCR Buffer for KOD FX Neo, 2 µL of 2 mM dNTPs, 10 pmole of each primer, 3 µL of eDNA solution and 0.4 U of KOD FX Neo. Thermal cycling was performed using two-step PCR cycling: 95 °C for 3 min followed by 35 cycles of 98 °C for 10 s and 65 °C for 30 s. Following PCR amplification, independent digestion using two restrictions enzymes (SpeI and HphI) and electrophoresis were conducted in the same manner as *Igawa et al. (2015)*. Then, 5 µl of the original PCR product or 15 µl of each PCR product digested with the two restriction enzymes were electrophoresed on a 2% agarose gel for 30 min at 100 V and visualized. We performed the experiment several times to confirm the reproducibility.

For *Rana japonica*, the amplified fragments are expected to have different digestion patterns between the western (including Hiroshima) and northern part (including Fukushima) of the Japanese mainland (*Igawa et al., 2015*). In all *Rana japonica* populations, amplicons digested with SpeI result in 255 and 259 bp subfragments. However, after Hph1 digestion, *Rana japonica* frogs from the western part have no subfragments, while those from the northern part have 301 and 220 bp subfragments (Fig. 2A). For *Rana ornativentris* and *Rana tagoi tagoi*, amplified fragments are commonly digested only with HphI, resulting in 231, 199, and 86 bp subfragments and 313 and 200 bp subfragments, respectively (Fig. 2A). To confirm the specificity of eDNA detection by our PCR-RFLP method, PCR products of sample no. 1, 10, 11, 14, 17, 19, and 20 were directly sequenced using the same primers and BigDye Terminator ver 3.1 (Life Technologies, Carlsbad, CA, USA) after PEG precipitation. The obtained nucleotide sequences were annotated by NCBI blastn (https://blast.ncbi.nlm.nih.gov).

## qPCR assay

To compare sensitivity between the methods, we conducted real-time qPCR assay for samples from Fukushima to compare the sensitivity of eDNA detection for *Rana tagoi*

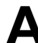

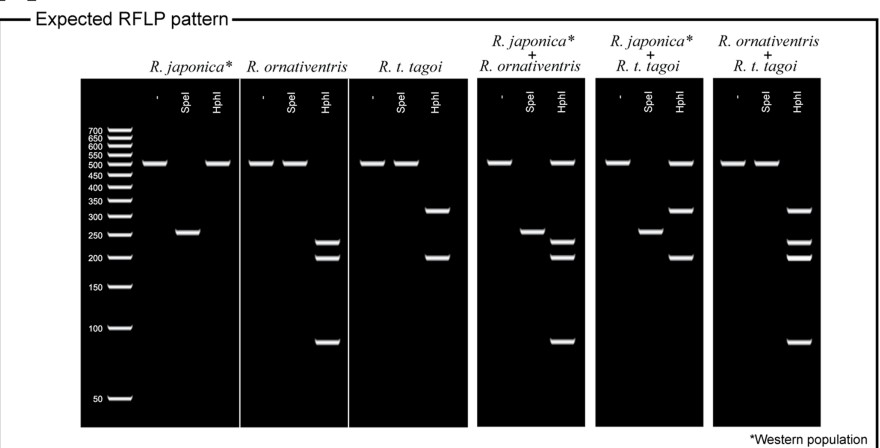

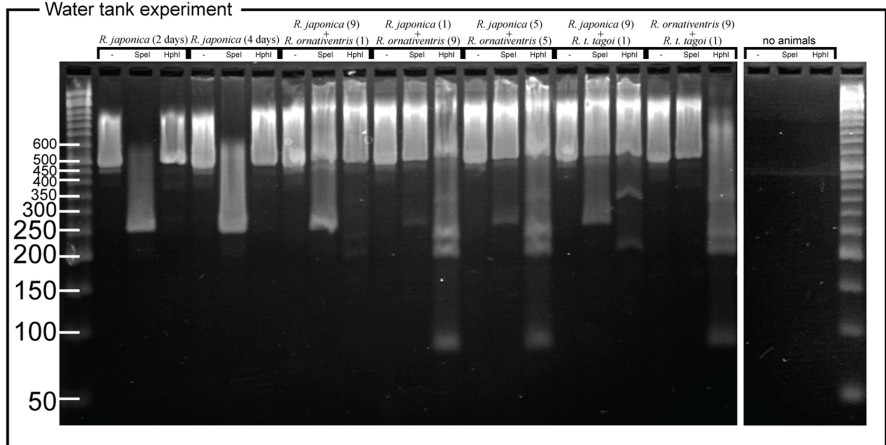

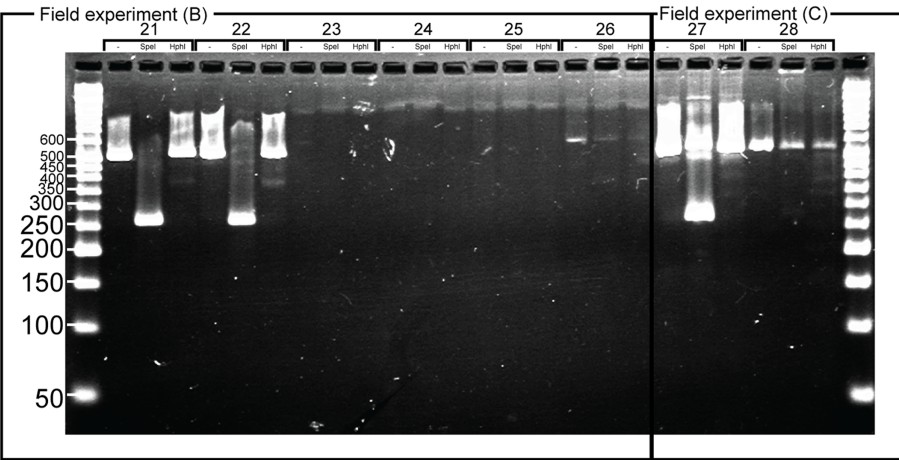

**Figure 2 PCR-RFLP and gel electrophoresis expected banding patterns (A) and results of water tank experiment (B) and field experiment (C).**

*tagoi*. At first, we defined regions to design the primers and TaqMan MGB probes for the qPCR assays for this species using PrimerExpress 3.0.1 (Thermo Fisher Scientific, Carlsbad, CA, USA), for amplification of 106 bp fragments: Rtag16SrRNA-F, 5′-AGAAGGAACTCGGCAAACCTT-3′; Rtag16SrRNA-R, 5′-CCGCGGCCGTTGAAT-3′; Rtag16SrRNA-Pr, 5′-[FAM]-CCAGCCTGCCCAGTG-[NFQ]-[MGB]-3′. To prove that the primers did not amplify other sympatric frog species, we confirmed no amplification when the primers were applied to DNA extracted from *Rana japonica* and *Rana ornativentris* tissues. However, in addition to *Rana tagoi tagoi*, three closely related species (i.e., *Rana sakuraii*, *Rana tagoi okiensis*, and *Rana tagoi yakushimensis*) were also detected in an in silico specificity screen, which was performed using Primer-BLAST, with the setting "nr" for database and "vertebrates" for organisms parameters (http://www.ncbi.nlm.nih.gov/tools/primer-blast/). *Rana sakuraii* may inhabit sympatrically with *Rana tagoi tagoi* in our field survey. In contrast, *Rana tagoi okiensis* and *Rana tagoi yakushimensis* never share distribution with our target species because they are only found on Oki island in Shimane Prefecture and the Yakushima islands in Kagoshima Prefecture, respectively. Thus, in this study, the qPCR primers we developed are specific for *Rana tagoi tagoi*/*Rana sakuraii*.

Environmental DNA was quantified using a StepOnePlus™ Real-Time PCR system (Life Technologies, Carlsbad, CA, USA). Each TaqMan reaction contained 10 μL of TaqMan® Environmental Master Mix 2.0 (Thermo Fisher Scientific, Carlsbad, CA, USA), 1 μL of the primer (900 nM)/probe (125 nM) mix, 7 μL of distilled water, and 2 μL of eDNA extract. The PCR cycles were as follows: 2 min at 50 °C, 10 min at 95 °C, then 55 cycles of 15 s at 95 °C, and 60 s at 60 °C. In order to produce standard DNA for the qPCR, a target amplicon was inserted into a pMD20-T vector (Takara Bio, Shiga, Japan), and the vector was digested with BamHI. A dilution series of the plasmid containing $1 \times 10^{1}$ to $1 \times 10^{4}$ copies was amplified as standards in duplicates in all qPCR assays. The qPCR for each sample was performed in eight wells, and the mean was used as the concentration of eDNA (copies/L). If any of the eight replicates of each sample yielded a positive result, the sample was designated as containing *Rana tagoi tagoi*/*Rana sakuraii* eDNA. As a negative control, each qPCR assay included eight wells that contained no template (2 μL of DNA-free water) (i.e., no template control (NTC)). In addition, using the primer-probe set of this species, qPCR amplicons were sequenced directly from a positive PCR of the field samples ($N = 6$) after treatment with ExoSAP-IT (USB Corporation, Cleveland, OH, USA) or DNA fragment isolation with FastGene™ Gel/PCR Extraction Kit (NIPPON Genetics, Tokyo, Japan) following agarose gel electrophoresis. Products were sequenced by a commercial sequencing service (Takara Bio, Shiga, Japan) or by our experiment using ABI3130xl automated sequencer (Thermo Fisher Scientific, Carlsbad, CA, USA).

The range of qPCR efficiency across the entire study, calculated from the slope of standard curves, was 91.937–94.716%, and the $R^2$ value for standard curve was 0.997. Since we could detect two copies of DNA in at least one of the three replicates, we defined the limit of detection (LOD, the lowest concentration where all triplicate samples registered a Ct value) for DNA from *Rana tagoi tagoi* using qPCR assay as two copies.

## RESULTS

### Water tank experiments

For the eDNA solutions from water tank experiments, we successfully amplified 16S rRNA fragments from all samples, each showing a single band following gel electrophoresis (Fig. 2B). The band patterns of digested fragments in each sample were completely matched with the species that were held in each tank, except for the negative control tank without animals (Fig. 2B). Notably, for the samples from water tanks in which two species were cultured, we could simultaneously amplify eDNA from both species whereby both band patterns specific to two species were displayed.

### Field survey and experiments

In the survey in Fukushima (A), field observation of *Rana tagoi tagoi* adults were recorded in 15 of 20 sites (Fig. 1; Table S1). Among the sites where *Rana tagoi tagoi* adults were observed, adults of *Rana ornativentris* were also observed sympatrically in five sites (Table S1). However, we could not identify presence of either *Rana tagoi tagoi* or *Rana ornativentris* by PCR-RFLP analyses of eDNA from any of the samples from Fukushima (Fig. S1). Subsequently, we applied the more sensitive *Rana tagoi tagoi/ Rana sakuraii*-specific qPCR which could still only confirm presence of *Rana tagoi tago*i in five of the 15 sites with field observations of adults of the species (Fig. 1; Table S1). Our direct sequences of the qPCR amplicons (106 bp product size) were almost identical to haplotypes of *Rana tagoi tagoi* deposited in DNA Data Bank of Japan (DDBJ). In addition, no amplification was observed in any control samples (i.e., filtering controls and NTC).

   We could obtain clear sequence data of PCR products from site no. 1 and 21, while the others showed mixed chromatograms that could not be further interpreted. In Fukushima site no. 1, we also reported field observation of *Bufo japonicus* tadpoles, a species outside of our target species. PCR amplification of eDNA from the same site resulted in an approximately 600 bp fragment; subsequent sequencing and BLAST search confirmed a match of nucleotide sequence with partial sequence of *B. japonicus* (= *B. j. formosus*) 16S rRNA (AB159565, a haplotype from Atsumi, Yamagata Pref. (*Igawa et al., 2006*)).

   From field experiment (B) and (C), field observation of *Rana japonica* tadpoles was recorded for three sites (no. 21, 22, and 27; Fig. 1; Table S1). This was consistent with our PCR-RFLP which showed *Rana japonica*-specific banding patterns following SpeI restriction digestion in the same samples (Fig. 2C). We also confirmed the sequence from site no. 21 had BLAST identity with partial sequence of 16s rRNA from *Rana japonica* (LC014155, a haplotype from Hiroshima University (*Igawa et al., 2015*)). In addition, PCR-RFLP analyses of site no. 28 supports the presence of *Rana japonica*, despite no field observation of tadpoles at this small drainage site. We also confirmed amplification of a fragment that may be amplified from non-target species in site no. 21, 22, 27, and 28. These fragments were digested with HphI and showed weak bands around 400 and/or 300 bp that were different from the target species.
## DISCUSSION

As shown in our water tank experiment, eDNA of *Rana* species was successfully amplified and we could even simultaneously identify each pair of the three species using PCR-RFLP method. Following success of this newly developed protocol, we applied it to a variety of field collected water samples.

In field experiment (A) at Fukushima, despite frogs being identified in 15 of 20 sites by field observation, we could not detect eDNA of *Rana tagoi tagoi* or *Rana ornativentris* using our PCR-RFLP method. Even when we applied the more sensitive qPCR approach for detecting *Rana tagoi tagoi*, this method only confirmed presence of the species in five of the sites. There are differences in detection sensitivity across PCR-RFLP and qPCR methods: qPCR is rather sensitive compared to conventional PCR in this case. In addition, the more expensive approaches like digital PCR and NGS of eDNA could prove to be more sensitive, despite the hurdles associated with cost.

The lack of detection by PCR-RFLP and limited sensitivity by qPCR could be attributed to the water system of the environment. Such factors like water volume, water discharge and flow rate could markedly impact the concentration of eDNA. It seems that flowing waters of lotic habitats, such as in rivers or streams, could have lower eDNA concentrations due to dilution from fresh water upstream. In particular, while only 500 mL water samples were collected in this study, previous studies extracted eDNA from more than one L of water (*Pilliod et al., 2013*; *Baldigo et al., 2017*). Such difference in water sampling volume might be associated with the lack of eDNA detection. In the future, increasing the volume of water collection might improve eDNA detection power.

In addition, *Rana tagoi tagoi* adults and their spawned tadpoles utilize small torrents or subsoil water systems only in the breeding season. According to *Maeda & Matsui (1999)*, *Rana tagoi tagoi* breed during late June in colder regions of Japan like Fukushima; our sample sessions occurred at both before and after this breeding period and thus could result in low level of eDNA in the water. Overall, the detection of *Rana* species from river water samples could potentially be challenging compared to static water samples, and we need to select the appropriate time of sampling considering ecology and behavior of the target species. Nevertheless, if future optimization can overcome these limitations, there is potential to apply such approaches to effectively monitor distribution of frogs in regenerating habitats in vicinity to the FNPP.

We also applied PCR-RFLP to static water samples collected in the field. From the eight sites in field experiments (B) and (C), PCR-RFLP could successfully detect eDNA in all three sites where *Rana japonica* tadpoles were visually observed. An additional site, site no. 28, had no tadpoles observed in field study, but showed some evidence of presence from PCR-RFLP analyses. This site is a small drainage site with a small volume and water flowing upstream from a paddy field. Thus, it seems possible that PCR-RFLP can detect presence of frogs that may inhabit upstream or in close proximity, even in the absence of visual confirmation. When we compare this site of flowing water to that of sites in Fukushima, the water volumes and flow rate are much smaller and slower, respectively.

## CONCLUSION

We have successfully developed a cost-effective method for detecting three common *Rana* frog species from Japan. This approach works well in field-collected samples that originate from static or slow-flowing water systems. However, our approach cannot be applicable to lotic water systems. Further exploration of alternative cost-effective methods are required, and could include collection of larger volumes of water and experimental assessment of the impact of various factors like water volume and flow rate on eDNA concentration. Nevertheless, this method has potential to be used to monitor frog populations that are facing decline due to urbanization, especially *Rana japonica* which inhabits static water bodies like rice paddy fields. Our approach also has potential to be adapted and optimized for detection of other amphibian species from field sites.

## ACKNOWLEDGEMENTS

We thank the Department of Molecular and Functional Genomics, Interdisciplinary Center for Science Research, Organization for Research and Academic Information, Shimane University for providing the DNA laboratory facilities. All methods adhered to the laws of Japan.

### Funding

This work was supported by a Scientific Research Grant from the Kurita Water, Japan and Environment Foundation Grant, by the Environment Research and Technology Development Fund of the Ministry of the Environment (4RF-1302), and Environmental Restoration and Conservation Agency (4-1503 and 4-1602), Japan, and the faculty of Life and Environmental Science in Shimane University, Japan provided financial support towards publishing this study. The funders had no role in study design, data collection and analysis, decision to publish, or preparation of the manuscript.

### Grant Disclosures

The following grant information was disclosed by the authors:
Scientific Research Grant from the Kurita Water, Japan.
The Environment Research and Technology Development Fund of the Ministry of the Environment (4RF-1302), and Environmental Restoration and Conservation Agency (4-1503 and 4-1602), Japan.
Faculty of Life and Environmental Science in Shimane University, Japan.

### Competing Interests

The authors declare that they have no competing interests.

### Author Contributions

- Takeshi Igawa conceived and designed the experiments, performed the experiments, analyzed the data, contributed reagents/materials/analysis tools, prepared figures and/or tables, authored or reviewed drafts of the paper, approved the final draft.

- Teruhiko Takahara conceived and designed the experiments, performed the experiments, analyzed the data, contributed reagents/materials/analysis tools, authored or reviewed drafts of the paper, approved the final draft.
- Quintin Lau performed the experiments, authored or reviewed drafts of the paper, approved the final draft.
- Shohei Komaki performed the experiments, approved the final draft.

### Animal Ethics

The following information was supplied relating to ethical approvals (i.e., approving body and any reference numbers):

Hiroshima University Animal Research Committee provided full approval for this research (Approval number: G17-4).

### Field Study Permissions

The following information was supplied relating to field study approvals (i.e., approving body and any reference numbers):

Permission to enter the survey area was obtained from the Iitate Vilage (approval number: 2705-01 and 2707-0) and Namine Town (approval number: 755 and 1959).

### Data Availability

The raw electrophoresis images are available in the Supplemental Files.

### Supplemental Information

Supplemental information for this article can be found online at http://dx.doi.org/10.7717/peerj.7597#supplemental-information.

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
