# Peer review of "An application of PCR-RFLP species identification assay for environmental DNA detection"

_PeerJ, doi:10.7717/peerj.7597_

## Round 0.1 · original submission · Major Revisions

We have now heard back from 2 expert referees, both of whom are supportive of your work, but each of which also have several comments and suggestions for improvement prior to publication. My own reading supports the feedback of the referees, and I recommend a major revision of the manuscript. In particular I find myself in agreement with both that the combination of lab and field trials, validation of positives by direct observation, and direct sequencing of amplicons are all strengths of the paper and that the paper is likely to become acceptable following some revisions to address the issues raised by the reviews.

The particular concerns with inconsistency of bands in the gel images and the lack of a laboratory trial with all 3 species should be addressed. My expectation is that the gel image included in the manuscript is one of the best you had, so I agree with the referee that it would likely be difficult to score if the average gel quality was much lower than these. Likewise, I agree with point 5 raised by the second referee – this is all about DNA concentration and persistence in the field. I don’t know why we would expect any real differences between lentic and lotic habitats other than dilution in flowing water, so either the authors ought to explain why they feel differently, or the results should be discussed in that context.

Beyond that, I expect the comments will be relatively simple to address. Please ensure that you address each of the points raised by the referees in your revision and include the details of how you have responded to each in your rebuttal letter. Although these revisions are not extensive, given the request for an additional trial (the 3 species comparison) and better-quality gels, I consider this a major revision. I expect it should be relatively straightforward to add these components however, and I look forward to seeing the revised manuscript.

Reviewer 1 ·

Basic reporting

On the whole, the manuscript is well constructed and presents valuable and interesting results. Some refinements will improve its legibility and the presentation of the data.

Specific suggestions for introduction:

• Introduction
o On line 65&66, the authors state that the PCR method has not been applied to other terrestrial species, however an article cited earlier (Olson et al 2013) does so with salamanders and Piaggo et al, 2013 develops a PCR assay to detect pythons in Flordia waters.
 Citation: Piaggio, A. J., R. M. Engeman, M. W. Hopken, J. S. Humphrey, K. L. Keacher, W. E. Bruce, and M. L. Avery. 2013. Detecting an elusive invasive species: a diagnostic PCR to detect Burmese python in Florida waters and an assessment of persistence of environmental DNA. Molecular Ecology Resources 14:374–380.
o 80-90: The section describing the ecological relevance and importance of the target species to conservation may be more effective if presented earlier in the text, before the details about life history and distribution.
o 99-101: The authors describe sampling near the Fukushima power plant, but it’s not entirely clear whether this is relevant to the study beyond it being simply a place they looked for frogs. Is there some connection to the Fukushima disaster?
o Did they sample in spring/summer for any particular reason?


Specific suggestions for results section:

• Results
o For the aid of the reader, it might be helpful to provide examples of the banding patterns expected from different combinations of frog species.
o Supporting Table 1 appears to be missing from the supporting information
o Were the sequences generated in this study deposited in GenBank? If so, please provide accession numbers.
o 265-275: It would be very helpful to see a visual summary (figure or table) of what was detected by which methods in both the laboratory and field studies. This is currently presented as +/- on the maps, but I think it would be nice to see presented on its own. That will make it easier for the readier to assimilate the results of the study.
o 276&277: Was Bufo observed visually and in the eDNA results? Was this sequence generated from the eDNA PCR product? I think it was, but it could be described a bit more clearly.



• Language/grammar: overall, the English is well-constructed and legible, but the manuscript might benefit from another read-through and perhaps a review by a native speaker to resolve some awkward sentences and grammatical inconsistencies. Here are a few specific recommendations:
o 42&42: consider something like “the ecology of species and their ecosystems”
o 45&46: instead of “sympatric related species,” try “closely related sympatric species”
o 54&55: I suggest something like “This potentially places limits on the number of institutions able to conduct eDNA studies”
o 55-57: “For ensuring sustainability of monitoring activities and conservation of species diversity, an efficient method should be developed for easy application in many laboratories close to a field site.” This sentence is cumbersome and should be reworded.
o 79: Begin sentences with the full genus & species names
o 82: “Breau” should be “bureau”
o 83: “Depertmet” should be “Department”
o 101: Use “effectiveness” instead of “effectivity”
o 135&136: “For exemplification of applicability in various scales of distribution monitoring studies” is also quite awkward
o 145-147: I suggest something like: “To prevent contamination by the field sampler, we began collecting at downstream sites, and moved upstream as subsequent samples were collected.”
o 148: remove “immediately”
o 149-150: instead of “presence and non-detection,” use “presence or absence”
o 160: instead of “particle retention,” use “mesh size”

Experimental design

The experimental design of this study largely falls within the desired criteria. Here are some specific suggestions regarding the methods:
• Methods
o 112: the negative control is described as water previously used for other non-target frog species. Was this done (as opposed to using completely clean water) to test for non-specificity of their primers? If so, it may help to add a sentence explaining that.
o 123 & 141: why did the authors choose to filter only 500 mL? Other eDNA studies seem to filter 1L at minimum. Was there any replication? This may be one of the reasons for the lack of detection in running water.
o 123: How were the DNA-free bottles sterilized?
o 124 & 148: How long were the samples kept frozen before they were filtered?
o 133-141: I think it would help to have a broader map of Japan as a whole upon which to place the more specific sampling-location maps. As a non-Japanese reader, it would be helpful to place the study in a geographic context. Similarly, it would be helpful to have the sample locations presented in a table.
o 143: Is it necessary to give the date and time of sampling? If there’s some relevance to the life history or ecology of the target species, please provide that. Otherwise, it’s probably extra detail that can be cut out.
o 149: This section is very short, but it might be helpful to place the visual survey methods under their own sub-heading.
o 159: What was the reasoning behind the choice of 0.7 um mesh size for the filters? There’s nothing explicitly wrong with that but many other eDNA studies seem to use 0.2-0.45 um filters.
o 160: Please describe the sterilization process in a bit more detail: what was rinsed and for how long?
o 161: Were the filtering controls carried to the sample sites? Or were they just laboratory controls?
o 163: Was the aluminum foil sterilized?
o 164-175: What sort of contamination controls were used during DNA extraction (surface sterilization, laminar flow hood etc.)?
o 197: Is there a citation for the stated expectation of different digestion patterns?
o 203-207: It’s not clear to me exactly what was sequenced? Was it the eDNA PCR products? Specifically, which amplified fragments were sequenced?
o 208-247: It might be helpful to have the qPCR section placed under its own sub-heading
o 248-251: The section on contamination avoidance could be moved to the top of the molecular methods section.

Validity of the findings

The findings of this study are presented clearly. There were some questions about the filter controls used, which I addressed in the experimental design section of this review. For lack of a better section in which to include it, here are some specific suggestions regarding the discussion section:

• Discussion
o 301: Is the Zemtsova, Montgomery & Levin, 2015 citation relevant to this statement? That study uses qPCR to detect pathogens in blood. Is there a citation that supports this claim for eDNA detections?
o 304-308: Many other studies have been successful at detecting target species in flowing water. It might be helpful to include discussion of what may explain the difference in results between those studies and this one. One possibility that stands out to me is the volume of water that was filtered. Other studies also chose to sample in eddies or pools in flowing watercourses, where genetic material might accumulate. Some also sample below the surface.

Additional comments

This study provides a valuable test of a tool that may simplify rapid ecological survey of an increasingly threatened group of animals. While they failed to detect the presence of frogs in flowing water, the PCR assay is shown to be useful for static water. Further experimentation and refinement may enable the assay to be used for flowing water habitats as well.

Reviewer 2 ·

Basic reporting

The aim of the study was to develop a PCR-RFLP method for eDNA detection of three rana frog species in Japan. The manuscript in general is well written and structured, thought there were several errors in the figure legends and text (see individual comments). The strengths of the study to me were 1) the combination of the lab and field trials and 2) the validation of the PCR-RFLP by field observations, qPCR, and sequencing of some of the amplicons.

Experimental design

3. In the methods, the authors had 8 replicates for qPCR but it seems there was only one replicate for conventional PCR. I wonder why and if this may be another reason for false negatives in the field trials. eDNA studies with conventional PCR typically had multiple replicates as far as I know.
4. There were no lab treatments with all three species. Don’t they coexist in the field at all? If so, does this method still work with all three species?

Validity of the findings

5. The conclusion that the PCR-RFLP works for lentic habitats but not lotic habitats is an interesting one. I wonder though if there were any other differences, other than eDNA concentration as discussed, between these two types of habitats that would affect eDNA detection. If the difference is only eDNA concentration as discussed, this means the method cannot detect eDNA even in lentic habitats with low eDNA concentrations or the method should be able to detect eDNA in lotic habitats with high eDNA concentrations.

Additional comments

The aim of the study was to develop a PCR-RFLP method for eDNA detection of three rana frog species in Japan. The manuscript in general is well written and structured, thought there were several errors in the figure legends and text (see individual comments). The results are inconclusive, which is fine according to one of the review criteria. The strengths of the study to me were 1) the combination of the lab and field trials and 2) the validation of the PCR-RFLP by field observations, qPCR, and sequencing of some of the amplicons. That said, I wonder about some issues/questions as follows. With appropriate responses and revisions, the manuscript should be acceptable. It is a neat study. Cost-effective detection of eDNA definitely helps. I hope my comments are helpful.

1. Although in Intro it is said that the PCR-RFLP method was developed to detect the three common rana frogs in Japan, the results (Fig. 1) seem to show those for only one species (R. tagoi). It would be nice to show the results for all three species.
2. I wish the quality of the gel images were better. For example, in Fig. 1A it was difficult to see the unique band (~250 bp) of R. japonica in the treatments in which there was only one R. japonica. The tagoi unique band (~300 bp) showed up pretty weak in all tagoi treatments. Then, the same faint bands seem to be present in the R. ornativentris treatments. In Fig 1B, there seem to be faint bands at ~400 bp length. Do they mean anything? I am sure that the authors were able to tell those better by looking at the actual gels. But with the images in the Fig. 2, it is difficult for readers to see them.
3. In the methods, the authors had 8 replicates for qPCR but it seems there was only one replicate for conventional PCR. I wonder why as eDNA studies with conventional PCR typically had multiple replicates as far as I know. This may be another reason for false negatives in the field trials.
4. There were no lab treatments with all three species. Don’t they coexist in the field at all? If so, does this method still work with all three species?
5. The conclusion that the PCR-RFLP works for lentic habitats but not lotic habitats is an interesting one. I wonder though if there were any other differences, other than eDNA concentration as discussed, between these two types of habitats that would affect eDNA detection. If the difference is only eDNA concentration as discussed, this means the method cannot detect eDNA even in lentic habitats with low eDNA concentrations or the method should be able to detect eDNA in lotic habitats with high eDNA concentrations.
6. In Abstract and Intro, the authors pitched the PCR-RFLP for amphibians in general when the method is only for the three Japanese rana species. Given the scope of the journal, the limited applicability is fine. But it is still misleading.

Individual Comments

Abstract
L 26: Insert “using quantitative PCR or next generation sequencing” after “the cost of the eDNA detection.” There are enough published studies that used traditional PCR for eDNA detection.
L 27-28: “foraying into” sounds odd for the use.
L 28-29: Unlike eDNA barcoding primers for amphibians (Valentini et al. 2016), the PCR-RFLP method discussed in this study is specific for those three Japanese rana species. I am not sure how the specific PCR-RFLP method benefits studies of amphibians in general
L 31: Inclusion of the Fukushima Nuclear Power Plant triggered my interest. Yet, there is no relevant discussion about this in the manuscript.
L 34-35: Please see my major comment on lentic vs. lotic habitats
L 37: “the golden standard of field observation” The point of using eDNA for many is that field observation is not golden for some species. If it is golden, I wonder why the authors became interested in developing this eDNA method. Also a field survey of pond breeding amphibians (calling in the breeding season and counting egg masses) is easy and time efficient.

Introduction
L 42: Change “a species” to “the species”
L 43: Change “conservation of a species” to “their conservation”
L43-46: The sentence can be improved. More importantly, these issues are not limited to field surveys but also relevant to eDNA-based surveys.
L 58: Change “more efficient” to “more cost-effective”
L 66: “to-date this method has not been applied to terrestrial organisms including amphibians” sounds misleading. The authors compared their amphibian study with the Clusa et al. 2017 that developed PCR-RFLP for fish. Some amphibians can be described as “terrestrial” but their larvae are aquatic and this study analyzed eDNA in water samples.
L 66-67: Clusa et al. 2017 developed the method for three trout species commonly stocked in North Hemisphere. Thus, their method has a broad applicability. The PCR-RFLP in this study is specifically designed for the three Japanese frogs and no applicability to other amphibians.
L 86: Delete “which are often”
L 97: It is nice to explain why having various spatial scales in sampling was important.
L 100: Change “regenerating habitats after the disaster” to “habitats recovering after the disaster”

MM
L 113: Please explain why the water that previously contained other frog species was used as negative controls and what those species were.
L 116: I am not familiar with Tahara (1974). I encourage the use of Gosner (1960) staging as it is much more commonly used.
L 120: Was the tap water aged?
L 123: Insert “according to the treatments” after “specific periods”
L 136: I am not sure how having “various scales” has anything to do with checking the performance and sensitivity of PCR-RFLP
L 138-141: A slight mismatch between these site IDs and the ones in Fig. 1
L 141: 500mL is a small volume. Is this an issue for not detecting eDNA?
L 146: Insert “a” before “field sampler”
L 162: Insert “cross” before “contamination”
L 186: Please specify the length of the 16S rRNA fragment amplified
L 208-: Nice to add an explanation as to why R. tagoi was selection for the qPCR assay
L 234-235: Were there any replicates for PCR-RFLP? If not, it would be nice to have replicates. Some studies that used conventional PCR for eDNA detection had replicates.
L 245-247: This explanation about LOD was unclear to me. I would appreciate clarification.

Results
L 257-258: Please see my major comment.
L 270: Do you have evidence for qPCR being more sensitive than PCR-RFLP?
L 276-280: Nice to better explain how this part about Bufo is related to the development of PCR-RFLP for the three Rana frogs.
L 282-286: Again, the site numbers do not correspond to those in Fig. 1. Fig. 1C says that R. japonica was detected in site 27.

Discussion
L 300-301: I wonder if there is a clear agreement on the sensitivity of qPCR over conventional PCR. For example, Garland et al. (2011, Diseases of Aquatic Organisms) reported that conventional PCR is as sensitive as qPCR. Thus, it may depend on the system.
L 304-306: Please see my major comment

Figure 1
There are no explanations on the panels (A, B, and C) of Figure 1 in the legend. Also it doesn’t say this is about detection of which species out of the three.

Figure 2
I would appreciate better quality of the gel images (please see my major comment). Also (A) is missing from the legend.

Supp Table 1 (L 268) was missing

---

## Round 0.2 · accepted · Accept

Your revised paper has been re-examined by one of the referees from your earlier review, and they are satisfied that you have addressed the concerns of both reviewers from the original submission. I agree with their recommendation and am happy to move your submission forward into production. Thank you for selecting PeerJ to publish your work.

Reviewer 1 ·

Basic reporting

Upon review of the revised and resubmitted manuscript, the authors appear to have satisfied the original comments and concerns of both myself and the second referee and I believe it warrants publication as it is.

Experimental design

See section 1.

Validity of the findings

See section 1.

Additional comments

See section 1.